# Zinc and Ferritin Levels and Their Associations with Functional Disorders and/or Thyroid Autoimmunity: A Population-Based Case–Control Study

**DOI:** 10.3390/ijms251810217

**Published:** 2024-09-23

**Authors:** Hernando Vargas-Uricoechea, Karen Urrego-Noguera, Hernando Vargas-Sierra, María Pinzón-Fernández

**Affiliations:** 1Metabolic Diseases Study Group, Department of Internal Medicine, Universidad del Cauca, Carrera 6 Nº 13N-50, Popayán 190001, Colombia; karenurrego@unicauca.edu.co (K.U.-N.); hernandod-vargass@unilibre.edu.co (H.V.-S.); 2Clinical Epidemiology Unit, Universidad Libre, Cali 760043, Colombia; 3Health Research Group, Department of Internal Medicine, Universidad del Cauca, Popayán 190003, Colombia; mpinzon@unicauca.edu.co

**Keywords:** zinc, ferritin, thyroid autoimmunity, thyroid function

## Abstract

Population zinc and iron status appear to be associated with an increased risk of thyroid function abnormalities and thyroid autoimmunity (AITD). In the present study, we aimed to determine whether zinc and/or iron levels (assessed by ferritin levels) were associated with the presence of AITD and with alterations in thyroid function. A population-based case–control study (*n* = 1048) was conducted (cases: *n* = 524; controls: *n* = 524). Participants were measured for blood concentrations of zinc and ferritin, TSH, FT4, FT3, and thyroid autoantibodies. No significant differences were found in relation to ferritin levels between cases and controls. Among cases, the prevalence of low zinc levels in those with hypothyroidism (both subclinical and overt) was 49.1% [odds ratio (OR) of low zinc levels: 5.926; 95% CI: 3.756–9.351]. The prevalence of low zinc levels in participants with hyperthyroidism (both subclinical and overt) was 37.5% [OR of low zinc levels: 3.683; 95% CI: 1.628–8.33]. The zinc value that best discriminated the highest frequency of AITD was 70.4 µg/dL [sensitivity: 0.947, 1–specificity: 0.655, specificity: 0.345]. The highest frequency of AITD was calculated based on a zinc value <70 µg/dL (relative to a normal value), with this frequency being significantly higher in cases than in controls [OR: 9.3; 95% CI: 6.1–14.3 (*p* = 0.001)]. In conclusion, the results of our study suggest that zinc deficiency is associated with an increased frequency of functional thyroid disorders and thyroid autoimmunity.

## 1. Introduction

Autoimmune thyroid disease (AITD) is the product of multiple environmental factors that act on a basis of genetic susceptibility and some epigenetic mechanisms. AITD has two extremes of clinical and/or biochemical presentation: hypothyroidism (in the context of Hashimoto’s thyroiditis—HT) and hyperthyroidism (in the case of Graves–Basedow Disease—GBD) [1].

In GBD, there is a loss of immune tolerance with infiltration of T lymphocytes in the thyroid, along with the activation of B lymphocytes and an increase in the synthesis and secretion of autoantibodies directed against the TSH receptor (TRAb) [1,2].

On the other hand, in HT, there is a cellular immune response (with high inflammatory load and apoptosis) with great tissue destruction. HT shares some humoral mechanisms with GBD, with the presence of autoantibodies against thyroid peroxidase (TPOAb) and against thyroglobulin (TgAb) [1,2,3].

The worldwide prevalence of GBD is 2% in women and 0.5% in men, while that of HT is 17.5% in women and 6.0% in men; despite its high frequency, the specific mortality rate for AITD is low [1,2,3].

However, the risk of mortality from any cause or from a cardiovascular origin is increased by 35–400% and 20%, respectively, in patients with manifest hyperthyroidism (compared to euthyroid individuals), and in cases of thyroid storm, mortality is between 3.5 and 17%. The triggering mechanism is not well established, although it can be explained by the increase in endothelial dysfunction and hypercoagulability that is usually associated with hyperthyroidism [2,3,4].

On the other hand, the mortality rate for HT (specifically in individuals with hypothyroidism) is increased by 14% (compared to healthy controls). This is probably due to the fact that hypothyroidism can induce endothelial dysfunction, arterial hypertension, atherogenic lipid profiles, and/or insulin resistance [5,6].

Among the environmental factors related to AITD, some micronutrients, such as iodine and selenium, stand out. However, several studies suggest that there are other related micronutrients, such as zinc and iron [7,8,9].

Zinc participates in the synthesis of both TSH and TSH-releasing hormone (TRH). In addition, zinc also acts as a cofactor of type I and II deiodinases, intervening in the synthesis of thyroid hormones (T4 and T3), and acts as a component of T3 nuclear receptors, thus being a determinant in the functioning of these hormones [10,11].

Zinc also intervenes in the functioning of thyroid transcription factor 2 (TTF-2), which stimulates the expression of genes that encode for the synthesis of TPO and Tg. Additionally, the deficiency of this micronutrient has been related to changes in the tissue structure of the thyroid, apoptosis, and a higher prevalence of TPOAb and TgAb [12,13].

However, studies that have evaluated the effect of zinc supplementation (in relation to its possible effects on thyroid function and TPOAb and TgAb levels) are inconsistent, which has led to uncertainty regarding its usefulness and clinical recommendations [12,13,14].

Another essential trace element necessary for the biosynthesis of thyroid hormones and proper thyroid function is iron. The largest amount of iron in the body is found mainly in hemoglobin and myoglobin, but it is also found in other proteins, such as TPO, which is a key enzyme in the biosynthesis of thyroid hormones [14,15].

Severe iron deficiency (assessed by ferritin levels) can affect TPO activity, thyroid hormone synthesis, and the peripheral conversion of T4 to T3, and it can increase TSH levels; additionally, iron deficiency has also been found to be associated with a higher prevalence of TPOAb positivity [15,16].

As with zinc, iron supplementation with the aim of regulating thyroid function (or modifying the autoimmune phenomenon of the gland) has not yet been demonstrated [14,15,16].

We hypothesize that altered zinc and/or ferritin levels could have an effect on thyroid function and the risk of autoimmunity towards the gland. In the present study, we aimed to determine whether zinc and/or iron levels (assessed by ferritin levels) were associated with the presence of AITD and with alterations in thyroid function.

## 2. Results

### 2.1. Participants Included in the Study

Adult participants (≥18 years) were recruited from the study population in a period from June 2018 to December 2023. All patients diagnosed with AITD, with or without functional alterations of the thyroid (euthyroidism, hypothyroidism, hyperthyroidism), were included as a “case” group (*n*= 524), and the participants (from the same population from which the cases were chosen) without AITD and with normal thyroid function were included as the “control” group (*n* = 524) (Figure 1).

### 2.2. Differences between Baseline Characteristics

No differences were found between the cases and controls in relation to variables such as age, sex, body mass index (BMI), SES, origin, and blood pressure (probably related to the fact that these variables matched between cases and controls).

Most of the subjects in the study were women, and about two-thirds of the participants belonged to a high SES. Likewise, the majority of individuals lived in urban areas.

The prevalence of goiter was significantly higher in the cases than in the controls (OR: 11.4; CI 95%: 8.3–15.6), *p* = 0.000, and higher in women than in men (in both cases and controls; *p* = 0.032 and 0.041, respectively).

Among the cases, the prevalence of normal thyroid function was significantly lower than in controls (*p* = 0.000). Similarly, among the cases, the prevalence of subclinical thyroid dysfunction was significantly higher than overt thyroid dysfunction (*p* = 0.001 and *p* = 0.000 for hypothyroidism and hyperthyroidism, respectively). It was also found that the prevalence of subclinical thyroid dysfunction (among cases) was significantly higher in women than in men (*p* = 0.001 and *p* = 0.000 for subclinical hypothyroidism and subclinical hyperthyroidism, respectively).

No significant differences were found between cases in terms of the positivity of the three thyroid autoantibodies in relation to sex (Table 1).

### 2.3. Ferritin and Zinc Levels and Their Associations with Alterations in Thyroid Function

The median and interquartile range (IQR) values of ferritin between the cases and controls were similar, with no significant differences found (*p* = 0.278).

Furthermore, the zinc levels were significantly lower in the cases (*p* = 0.000), as was the proportion of subjects with zinc levels <70 µg/dL in this group of participants (*p* = 0.000) (Table 2).

Among the cases, the prevalence of low zinc levels in those with hypothyroidism (both subclinical and overt) was 49.1% (95% CI: 43.1–55.1), and the OR of low zinc levels (relative to the group of cases with normal thyroid function) was 5.926 (95% CI: 3.756–9.351). Meanwhile, the prevalence of low zinc levels in participants with hyperthyroidism (both subclinical and overt) was 37.5% (95% CI: 19.2–55.8), and the OR of low zinc levels (relative to the group of cases with normal thyroid function) was 3.683 (95% CI: 1.628–8.33).

### 2.4. Zinc Levels and Risk of Thyroid Autoimmunity

A complementary analysis was performed by performing a ROC curve analysis, evaluating different cut-off points in zinc levels (area under the curve—AUC) in order to determine which one best discriminated the highest frequency of AITD [AUC: 0.673 (95% CI: 0.640–0.706)] (Figure 2).

The zinc value that best discriminated the highest frequency of AITD was 70.4 µg/dL [sensitivity: 0.947, 1–specificity: 0.655, specificity: 0.345 (Youden index: 0.292)]. According to the above, the highest frequency of AITD was calculated based on a zinc value <70 µg/dL (relative to a normal value), with this frequency being significantly higher in cases than in controls [OR: 9.3; 95% CI: 6.1–14.3 (*p* = 0.001)].

Finally, in the logistic regression analysis (multivariate), the risk of AITD among those with low zinc levels (compared to those with normal values) was significantly higher in the cases [adjusted OR: 6.9; 95% CI: 4.4–10.9 (*p* = 0.000); Wald’s test: 69.461].

## 3. Discussion

In this study, we evaluated the possible associations between zinc and ferritin levels on the presence of AITD and/or functional thyroid disorders. However, the ferritin levels between cases and controls were very similar, and we found no differences between the groups.

This finding contrasts with previous studies and recent meta-analyses that conclude that iron deficiency (assessed by ferritin levels) is associated with alterations in TSH, FT4, and FT3 levels, as well as with TPOAb and TgAb positivity [17,18,19].

Nevertheless, these studies were mainly conducted in women (most of them in pregnant women), and several of them compared participants with iron deficiency (vs. healthy controls), with different definitions in relation to the cut-off points of ferritin levels (<12, <15, <20 ng/dL). Therefore, the failure to identify differences in the ferritin levels between the two groups evaluated can be explained, at least in part, by the fact that we excluded pregnant women, lactating women, women with a history of anemia, and women with abnormal uterine bleeding.

Thereby, this finding in our population does not rule out the possibility that iron has a role in the pathogenesis of AITD and in thyroid function, so it could be suggested that the effects of iron on the thyroid are especially reflected in populations with an established deficiency of this micronutrient and not in those with a normal nutritional iron status.

Among the participants (specifically among cases and in women), the higher prevalence of goiter and subclinical thyroid disease reflects the findings in previous studies, where a high population frequency of disorders associated with iodine intake and a probable effect associated with environmental goitrogens have been demonstrated (without population selenium deficiency being documented) [20,21].

However, in this study, we found a significantly high prevalence of zinc deficiency in the cases (and especially among those with hypothyroidism or hyperthyroidism compared to those who were euthyroid), contrasting with previously published studies reporting that low zinc diets (or low circulating levels) can cause certain changes in the structure of the thyroid gland and/or in the metabolism of thyroid hormones [7,8,9,22].

The possible association between low zinc levels and the presence of hypothyroidism in our population can be explained, in part, by a process established at multiple levels; e.g., zinc is involved in the synthesis of TRH and TSH through a process that is regulated by carboxypeptidase (which is a zinc-dependent enzyme) [23,24,25].

Therefore, the deficiency of this micronutrient can affect the pituitary–hypophysis-thyroid axis (at the level of the central nervous system).

Additionally, since zinc is a cofactor of deiodinases I and II, its deficiency can affect the conversion of T4 to T3 [26]. In addition, nuclear receptors for T3 contain zinc ions (hence, a zinc deficiency can be associated with a low conversion of T4 to T3 and with alterations in the binding of T3 to its nuclear receptor) [26,27,28].

It has also been reported that zinc deficiency induces structural tissue changes (thyroid) and increased apoptosis; moreover, zinc is necessary for the proper functioning of transcription factors such as TTF-2, and TTF-2 intervenes in the expression of genes that code for the synthesis of Tg and TPO [28,29,30].

Consequently, it could be proposed (hypothetically) that zinc deficiency is predisposed to the expression of genes involved in the synthesis of Tg and TPO, inducing these thyroid autoantigens to have a configuration that confers greater “antigenicity” and, eventually, a greater probability of triggering an autoantibody-mediated response (TgAb and TPOAb). The above could explain (at least partially) the association found between zinc deficiency and a higher frequency of AITD in our population.

Furthermore, the association identified between zinc deficiency and hyperthyroidism in our study is intriguing and differs from other studies with inconsistent results (both in rodents and in humans), where it has been documented that it is the excess (and not the deficiency) of zinc that may be associated with hyperthyroidism; in addition, several studies previously found that patients with hyperthyroidism had a high prevalence of low erythrocyte zinc levels [31,32,33,34,35].

In this sense, several hypotheses can be put forward. For example, some studies (in animals) have shown that zinc deficiency is associated with a significant increase in leptin levels. Likewise, in humans, zinc deficiency has been linked to low leptin levels and, in turn, hyperthyroidism; in fact, some studies have shown that low leptin levels are associated with hyperthyroidism (GBD) and that these levels tend to normalize once the hyperthyroidism is controlled [36,37].

The effects of zinc on leptin levels may be mediated by the increased activation of PPAR-ϒ (at both the mRNA and protein levels), the increased production of cytokines (such as IL-2 and TNF-α), the increased expression of leptin receptor mRNA at the hypothalamic level, and, finally, by the indirect stimulation of leptin synthesis (by increasing glucose consumption in adipose tissue) [36,37,38,39].

Furthermore, leptin has been shown to regulate (directly and indirectly) TRH production in the hypothalamus via the Janus kinase/signal transducers and activators of transcription (JAK/STAT) pathway. This increase in TRH release leads to increased TSH secretion (in the pituitary gland), stimulating thyrocyte function and proliferation [38,39]. However, concluding that there is a relationship between zinc deficiency, low leptin levels, and hyperthyroidism is not easy, since the mere presence of hyperthyroidism can induce a decrease in leptin levels [39,40,41].

Additionally, it should also be noted that zinc at the level of red blood cells is present mainly as a cofactor of carbonic anhydrase. In animal models, it has been found that T4 and T3 are capable of inhibiting the synthesis of the carbonic anhydrase isoenzyme B; therefore, the inhibition of the synthesis of this isoenzyme or alterations in its distribution or turnover (leading to a mild zinc deficiency) can be considered (hypothetically) as one of the causes explaining the low zinc levels in hyperthyroidism [42,43].

Finally, we found that a zinc level <70 µg/dL was associated with a higher frequency of AITD. To our knowledge, this is the first study evaluating a cut-off point for zinc levels and its association with AITD; therefore, considering that zinc deficiency is a highly prevalent condition worldwide (with a prevalence of 17–18%) [43,44], it could be suggested that the high prevalence of zinc deficiency could be one of the underlying causes of the increased susceptibility to AITD in the general population.

Therefore, a blood zinc level <70 µg/dL should be considered to evaluate alterations in thyroid function and in determining thyroid autoimmunity; however, intervention studies (supplementing zinc) have not been consistent with the majority of the thyroid outcomes evaluated (e.g., concentrations of TSH, FT4, FT3, TPOAb, and TgAb), reporting that the intervention does not seem to substantially modify these parameters. Therefore, zinc supplementation in this type of population is still controversial [7,11,45,46,47,48].

Our findings should be considered while recognizing their potential limitations; e.g., the characteristic of being a case–control study does not allow for a temporal sequence between exposure (specifically, zinc deficiency) and the outcome of interest (thyroid autoimmunity) to be precisely established.

Likewise, given the type of design used in our study, we were not able to establish a causal association and a definitive link between the zinc deficiency of our patients and the presence of hyperthyroidism.

Additionally, we matched the participants according to five potential confounders (origin, age, sex, BMI, and SES), which reduces the possibility of evaluating whether these characteristics are different in the cases and controls (with a potential increase in the risk of overmatching).

We could not exclude the impact of other potential confounding factors; e.g., we did not take into account nutritional habits related to the intake of foods containing zinc and iron, among others, which could have (eventually) influenced the differences found between the groups studied.

We cannot exclude the phenomenon of residual confounding factors (e.g., unreported or unrecognized intake of micronutrients or multivitamins) or the presence of endocrine disruptors that may potentially exacerbate or dilute the associations found.

Finally, further studies should evaluate the true role of ferritin in thyroid function and its association with thyroid autoimmunity (specifically in iron-deficient populations). In addition, the effect of zinc supplementation (in deficient individuals) on prespecified outcomes, such as the decreased risk of thyroid function disorders and their effect on thyroid autoantibodies, should be clarified by means of a robust clinical trial design.

## 4. Materials and Methods

### 4.1. Study Population

Between June 2018 and July 2024, a population study was carried out in the department of Cauca (Popayán, southwestern Colombia), with the objective of evaluating the distribution and behavior of AITD, as well as the frequency of thyroid functional status (euthyroidism, hypothyroidism—subclinical or overt, and hyperthyroidism—subclinical or overt), among other objectives and outcomes.

The study comprised individuals (civilian, non-institutionalized) from urban and rural areas who were invited to participate in the study through active visits to the different neighborhoods and communes of the Cauca department (Colombia).

For the global study, 17,430 individuals signed the informed consent form after the explanation of the purposes and objectives of the study and then filled out a questionnaire inquiring about previous health history (potentially eligible individuals); of these, 6011 were excluded for different reasons, finally resulting in 11,419 potential participants. Of these, 525 individuals were excluded (for various reasons), leaving 10,894 individuals (potentially eligible participants), from which the cases and controls were chosen.

Each case was assigned a control in a 1:1 ratio and matched according to possible confounding variables: origin (urban, rural), age, sex, BMI, and SES.

This information was recorded using the Windows Excel program (Microsoft, 2020) and processed with SPSS software version 25.0 (IBM-SPSS Inc., Chicago, IL, USA).

### 4.2. Study Design

A population-based case–control study.

### 4.3. Definition of AITD and Normal or Abnormal Thyroid Function

The definition of AITD was determined by measuring thyroid autoantibodies [TPOAb, TgAb, and anti-TSH receptor antibodies (TRAb)]; in this sense, the positivity of at least one of the three autoantibodies was defined as AITD [49].

Thyroid function was established by measuring thyrotropin (TSH) and free T4 (FT4); a normal TSH and FT4 value classified the individual as euthyroid; a high TSH value with normal FT4 made the diagnosis of subclinical hypothyroidism; a high TSH value with low FT4 identified the participant as having overt hypothyroidism; a low TSH value with elevated FT4 classified the individual as having overt hyperthyroidism; and a low TSH value with normal FT4 identified the individual as having subclinical hyperthyroidism. In these cases, free T3 (FT3) was additionally measured to rule out (or confirm) the diagnosis of hyperthyroidism due to T3 [50].

### 4.4. Exclusion Criteria

For the selection of both study groups, individuals with current use or use in the last three months of multivitamins (containing iodine/selenium/biotin/zinc/iron), amiodarone, lithium, or steroids; active smoking; previous treatment with radioactive iodine; thyroidectomy (partial or total); a history of irradiation to the head and neck; pregnancy or lactation; history of surgery or pituitary tumors; central hypothyroidism; history of anemia under treatment; abnormal uterine bleeding; chronic obstructive pulmonary disease (COPD); or chronic kidney disease (CKD) were excluded. Each of these exclusion factors was identified through patients’ self-reporting.

### 4.5. Anthropometric and Laboratory Measurements

Trained interviewers conducted face-to-face interviews to obtain a detailed picture of the participants’ sociodemographic data and medical history. Standard anthropometric data (age, height, weight, body mass index (BMI, in kg/m^2^), and blood pressure) were obtained from patients and controls. Additionally, the presence of goiter was determined by inspection and palpation (according to World Health Organization criteria) [51].

Basal blood samples were obtained between 7.00 and 9.00 after an overnight fast. All samples were obtained via venipuncture, and the blood was centrifuged, frozen at −80 °C, and stored (Martha Perdomo Specialized Clinical Laboratory, Popayán, Colombia). 

The variables were measured using a chemiluminescent immunoassay (IMMULITE^®^ 2000 Systems Analyzers; Siemens, Munich, Germany) [52].

The positive titers for TPOAb, TgAb, and TRAb were ≥8.0 IU/mL, ≥18 IU/mL, and >1.75 IU/mL, respectively (manufacturer cutoffs). The normal values for TSH, FT4, and FT3 were as follows: TSH: 0.4–4.5 µIU/mL, FT4: 0.89–1.76 ng/dL, FT3: 3.5–8.3 pmol/L. The coefficients of variation were 4.96%, 8.0%, and 5.3% for TPOAb, TgAb, and TRAb, respectively; and 4.6%, 6.4%, and 6.7% for TSH, FT4, and FT3, respectively [52].

Zinc values were evaluated by means of the atomic absorption spectrometry technique (with linearity of up to 1000 µg/L and sensitivity of 10 µg/L), and the normal value range (established by the manufacturer) was 70–150 µg/dL. Likewise, ferritin levels were determined using the one-step sandwich immunoenzymatic method with final detection via fluorescence (ELFA) (VIDAS^®^Ferritin). The normal reference range for ferritin was 21–278 ng/mL (coefficients of variation: 5%) [53,54].

### 4.6. Statistical Analysis

The number of subjects for the study was calculated based on previous studies (conducted in the same study population), which found that the prevalence of positivity for thyroid autoantibodies was as follows: 5.3% for TRAb; 10% for TgAb; and 18.2% for TPOAb. Likewise, for Colombia, it has been established that zinc deficiency in adults is greater than 50% and iron deficiency (determined by ferritin) is 38.8% [48].

Information was collected from 1048 participant records: 524 cases and 524 healthy controls. 

A quality control of the records was carried out, including a descriptive analysis of each variable, cleaning the data, and identifying missing values. The results for continuous variables were expressed as medians and interquartile ranges (IQRs). Otherwise, categorical variables were presented as relative frequencies (%).

Comparisons of continuous variables between two groups were made using the Mann–Whitney *U* test. A paired t-test or Wilcoxon signed-rank test was used to compare zinc and ferritin values between groups. Differences in the proportions of different patient groups were compared using the Chi-square test (*X*^2^).

To evaluate critical points of zinc and/or ferritin levels with the risk of AITD, a ROC (receiver operating characteristic curve) analysis was used. To determine the association between the zinc and/or ferritin levels and AITD, a multiple conditional logistic regression was carried out.

The statistical analysis of the distribution of the antibodies was carried out based on the exact binomial distribution. All analyses were two-sided and a *p*-value of ≤0.05 was considered statistically significant (with an established power of 90%).

## 5. Conclusions

In conclusion, our study suggests that zinc deficiency is associated with an increased frequency of both functional disorders and the presence of thyroid autoimmunity. These findings allow us to propose some hypotheses and explain (in part) the broad pathophysiological scenario surrounding AITD.

Future research should focus on determining whether zinc supplementation reduces the frequency of such thyroid functional disorders and AITD.

## Figures and Tables

**Figure 1 ijms-25-10217-f001:**
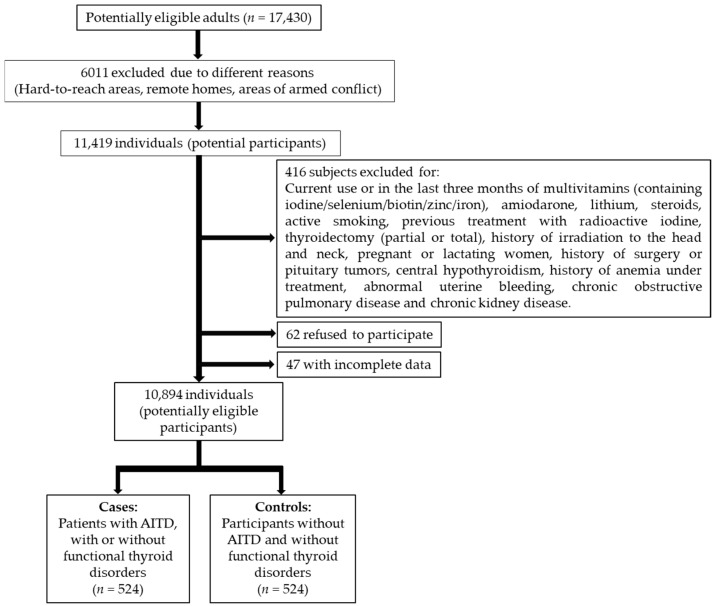
Flow chart of the participants included in the study.

**Figure 2 ijms-25-10217-f002:**
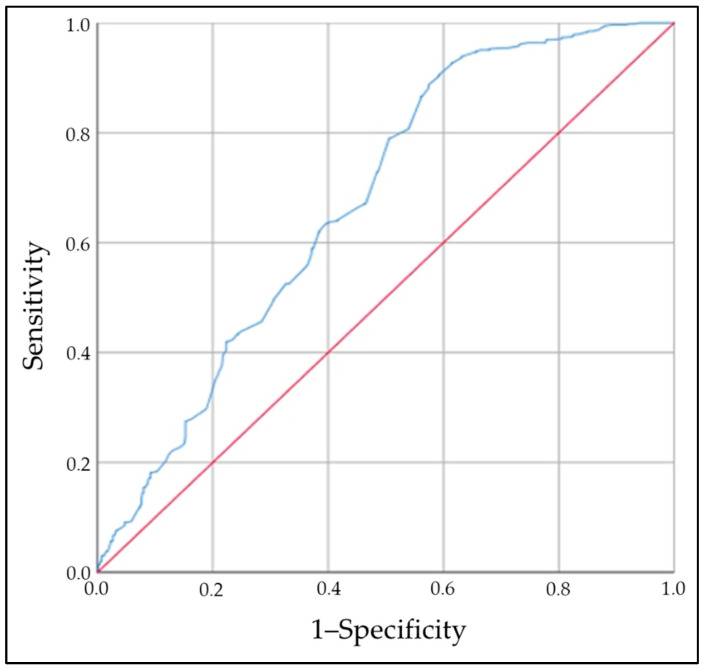
Area under the curve (AUC), which showed that the zinc value that best discriminated the risk of AITD was 70.4 µg/dL [AUC: 0.673 (95% CI: 0.640–0.706)].

**Table 1 ijms-25-10217-t001:** Anthropometric–sociodemographic characteristics and thyroid functioning status and positivity of thyroid autoantibodies in cases and controls.

Characteristics	Cases (*n* = 524)	Controls (*n* = 524)	*p*-Value
Age, median, and IQR	47.6 (33–64)	47.1 (34–63)	1.00
Female (%)	63.7	63.7	1.00
BMI, median, and IQR	27 (24.2–30.0)	27.5 (24.3–30.9)	0.387
Low–middle SES (%)	34.5	34.5	1.00
High SES (%)	65.5	65.5	1.00
Origin (urban/rural)	59.9/40.1	59.1/40.1	1.00
Systolic blood pressure (mmHg), median, and IQR	128 (111–147)	125 (109–143)	0.40
Diastolic blood pressure (mmHg), median, and IQR	79 (72–85)	77 (70–82)	0.39
Prevalence (%) of goiter (F/M)	60.9 (63.8/55.8)	12 (58.7/41.26)	0.000
Prevalence (%) of normal thyroid function	39.5	100	0.000
Prevalence (%) of subclinical/overt hypothyroidism (F/M)	33.6/20.8	NA	0.001
Prevalence (%) of subclinical/overt hyperthyroidism (F/M)	5.0/1.1	NA	0.000
Prevalence (%) of TPOAb positivity (F/M)	73.9 (73.1/75.3)	NA	0.41
Prevalence (%) of TgAb positivity (F/M)	49.6 (50.0/48.9)	NA	0.46
Prevalence (%) of TRAb positivity (F/M)	17.9 (19.2/15.8)	NA	0.38

Abbreviations: BMI: body mass index, F: female, IQR: interquartile range, M: male, NA: not applicable, SES: socioeconomic status, Tg: thyroglobulin, TPO: thyroid peroxidase, TgAb: anti-Tg antibodies, TPOAb: anti-TPO antibodies, TRAb: anti-thyrotropin receptor antibodies; TSH: thyrotropin.

**Table 2 ijms-25-10217-t002:** Distribution of zinc and ferritin values between cases and controls.

	Zinc Values (µg/dL)		*p*-Value	Ferritin Values (ng/mL)		*p*-Value
	Cases (*n* = 524)	Controls (*n* = 524)		Cases (*n* = 524)	Controls (*n* = 524)	
Median (IQR)	78 (61–91)	88 (78–103)	0.000	73.5 (32–125)	76 (39–148)	0.278
Low values (% participants)	34.5	5.3	0.001	20.6	18.1	0.31
Normal values (% participants)	63	88.9	0.001	76	79.4	0.12
High values (% participants)	2.5	5.7	0.08	3.4	2.5	0.15

## Data Availability

The data presented in this study are available on request from the corresponding author.

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
