# Peer review of "Zinc and Ferritin Levels and Their Associations with Functional Disorders and/or Thyroid Autoimmunity: A Population-Based Case–Control Study"

_ijms, 2024, doi:10.3390/ijms251810217_

Round 1
Reviewer 1 Report
Comments and Suggestions for Authors
The manuscript entitled “Zinc and Ferritin Levels and Their Associations with Functional Disorders and/or Thyroid Autoimmunity: A Population-Based Case–Control Study” analyzes the relationship between micronutrient-related parameters and thyroid disorders. The topic of the manuscript is within the scope of the journal and would be interesting to the readership involved in the research and/or management of thyroid and autoimmune disorder.
The manuscript is poorly structured, presentation of the results is not informative and clarifications are needed throughout the text. Major and minor issues need to be addressed (in order of appearance):
- The whole Introduction section needs to be rewritten. The background is too concise, while rephrasing is needed for the first two paragraphs (“some” is repeated several times). The exact novelty is not obvious and needs to be highlighted, since there are various previous studies analyzing the relationship between zinc and thyroid disorders. Background and the rationale for conducting the study should be elaborated. The final sentence of the Introduction is not adequate, since ferritin is not a micronutrient and the study does not explore the casual relationship between zinc/ferritin and thyroid disorders, just an association between them in a cross-sectional design.
- Except for the part describing the Figure 1, the whole segment of the Results up to section 2.5. belongs to Material and Methods. None of this data provides any information that is a result of a study. Paragraph 3 in 2.1: “11,419 participants” should be “11,419 potential participants”. Flow-chart in Figure 1 requires additional information: define the reasons for excluding 6011 participants.
- The first paragraph in 2.5. describes results which are expected, based on the study design. This should be highlighted, since the control group does not reflect the actual general population, being matched with cases for certain parameters.
- The Results are not informative in a current state. There should be a Table with data on zinc and ferritin levels for cases and controls segregated according to the exact diagnosis. “OR prevalence” is not an adequate term. It implicates that it describes the OR of hypo/hyperthyroidism in patients with low zinc. However, the data describes the OR of zinc deficiency in patients with specific thyroid disorder and the term should be “OR of low zinc”, or a similar term.
- In the first paragraph of 2.7., the results are poorly interpreted, since the cut-off does not associate with the highest risk of AITD, it presents with the highest discriminatory ability. There should be another Table with the data on zinc and ferritin levels for cases and controls segregated according to the AITD status. Term “risk” should be avoided, since there is no predictive significance of zinc and ferritin established through this type of study design.
- The conclusions are exaggerated and not well supported by the findings. The susceptibility toward thyroid dysfunction was not evaluated and the results are merely an indication of association between the analyzed parameters and traits.
Comments on the Quality of English Language
Minor editing is needed.
Author Response
Kind regards, attached to this message you will find the point-by-point answers to the concepts and comments made by reviewer 1.
Thank you very much for all the collaboration and support.

Reviewer 2 Report
Comments and Suggestions for Authors
This study aimed at evaluating the association between iron and zinc deficiency and risk of autoimmune thyroid disease and conditions of hyper- and hypothyroidism. Overall, the manuscript is well written, however some points need attention from the authors.
Abstract. Please correct the typo in “odds ratio (OR): 5.926; 95% CI: 3.756–9.351”.
Section 2. The authors should consider moving certain parts ( (2.1 2.2, 2.3, 2.4) to the Methods section. Only the final number of participants should be cited in the results.
Table 1. How did the authors calculate the p-value for variables such as the prevalence of clinical/subclinical hyperthyroidism/hypothyroidism and positivity for autoantibodies, since they may not be detectable in controls?
Section 2.6. I do not understand the double OR values for prevalence of low zinc values in controls (5.926 and 3.683).
Discussion. “Among the participants (especially in cases and in women), the higher prevalence of goiter and subclinical thyroid disease”. On the basis of what results can this be stated?
Discussion. “Consequently, it could be proposed (hypothetically) that zinc deficiency predisposes the expression of genes involved in the synthesis of Tg…”. For better understanding, please rephrase the text, possibly dividing it into two sentences.
Discussion. “Therefore, given the case–control design of our study, we were not able to establish a causal association and establish a definitive link between the zinc deficiency of our patients and the presence of hyperthyroidism.” This point is a limitation of the study. I suggest to insert a section only including strengths and limitations.
Conclusions. “In conclusion, our study suggests that zinc deficiency may increase susceptibility to both alterations in thyroid function and the presence of thyroid autoimmunity”. Please specify which alternations you are referring to.
Comments on the Quality of English LanguageModerate English revision
Author Response
Kind regards! Attached to this message you will find the point-by-point responses to the comments of reviewer 2, each of the modifications in the manuscript were highlighted in yellow. Thank you very much for all the collaboration and support! Hernando Vargas-Uricoechea. First author and corresponding author

Reviewer 3 Report
Comments and Suggestions for Authors
General Comment: The manuscript brings useful information regarding Zinc and Ferritin levels and their associations with functional disorders and/or thyroid autoimmunity. The authors investigated the possible associations of zinc and ferritin levels and hypothyroidism/hyperthyroidism and/or AITDT. The results are interesting. The authors showed that there are no significant differences in relation to ferritin levels between cases and controls. However, the authors claimed that zinc deficiency may increase susceptibility to both alterations in thyroid function and the presence of AITD. The reviewer believed that the present study is interesting. However, some concerns and questions must be addressed to clarify and improve the present version.
Specific comments:
Title
The title is informative and relevant to the major findings.
Abstract
In the abstract, the aim of the study should be mentioned clearly. Major results are properly presented. Number of participants should be mentioned in the abstract.
Introduction
The research gap/question is not clearly outlined. The introduction section does not provide sufficient background of the research topic. Authors should revise the introduction by describing the development mechanisms of Autoimmune thyroid disease (AITD). Please explain the causes/reason for the development of AITD. How many people around the world are suffering from AITD? How many people die annually due to AITD worldwide? If possible, please include this information to contextualize the importance of the study.
Results
In Figure 1: Is the potential participant's number, correct?
Subsections “2.2. Definition of AITD and Normal or Abnormal Thyroid Function”; “2.3. Exclusion Criteria”; “2.4. Anthropometric and Laboratory Measurements”, should be transferred in Materials and Methods section.
The results section is not well-organized and also not well-explained.
Discussion
The discussion section could be more focused. The discussion doesn’t properly reflect the results obtained in this study. Authors can give more emphasis to discussing their findings from multiple angles in this section.
Materials and Methods
In general, this section is not well described. Authors should provide sufficient detail about procedures and data so that the same procedures could be exactly repeated.
Conclusion
Major limitations and opportunities to inform future research are not addressed properly.
Overall comments: The manuscript is not well-written and not well-organized. Minor English changes are required.
Comments on the Quality of English LanguageMinor English changes are required.
Author Response
Kind regards! Attached to this message you will find the point-by-point responses to the comments of reviewer 3. Each of the changes made to the manuscript were highlighted in yellow. Many thanks for all the collaboration and support. Hernando Vargas-Uricoechea. First author and corresponding author.

Round 2
Reviewer 1 Report
Comments and Suggestions for Authors
The authors have made the required corrections and the manuscript is significantly improved.
Author Response
Kind regards, many thanks to the reviewer for each of the comments and suggestions on the manuscript. Cordially, Hernando Vargas-Uricoechea

Reviewer 3 Report
Comments and Suggestions for Authors
The revised version of the manuscript is improved but not satisfactory. The authors addressed my comments partially. They should give more emphasis on the introduction, results, and methods parts during revision.
Please find my specific comments below-
Introduction
The research gap/question is not clearly outlined. Authors should revise the introduction by describing the development mechanisms of Autoimmune thyroid disease (AITD). Please explain the causes/reason for the development of AITD. How many people around the world are suffering from AITD? How many people die annually due to AITD worldwide? If possible, please include this information to contextualize the importance of the study.
Results
In Figure 1: Is the potential participant's number correct? if there is 10894 eligible participants, why did the authors included only 524+524= 1048? Please explain.
The results section is not well-explained. Please cite the table and figure in the text where appropriate. Please mention the table or figure in the last sentence of the results. ("Finally, in the logistic regression analysis (multivariate), the risk of AITD among those with low zinc levels (compared to those with normal values) was significantly higher in the cases")
Discussion
The discussion section could be more focused. The discussion doesn’t properly reflect the results obtained in this study. Authors can give more emphasis to discussing their findings from multiple angles in this section. Please compare your findings with previously published reports.
Materials and Methods
In general, this section is not well described. Authors should provide sufficient detail about procedures and data so that the same procedures could be exactly repeated. If necessary, authors should use proper citations and references.
Conclusion
Major limitations are not addressed properly.
Author Response
Kind regards! Attached to this message you will find the point-by-point answers to the comments and suggestions of reviewer 3 (Round 2). additionally, attached also the English editing certificate (MDPI, Author services). Many thanks for all the collaboration and support, Cordially. Hernando Vargas-Uricoechea

Round 3
Reviewer 3 Report
Comments and Suggestions for Authors
No more comments, Best of Luck